# Comparison of methods for determining parameters related to ankle joint quasi-stiffness during quiet standing

Kaylie Lau[1,2], Kai Lon Fok[1,2], Jonguk Lee[1,2], Kei Masani[1,2]*

1 Institute of Biomedical Engineering, University of Toronto, Toronto, Ontario, Canada, 2 KITE - Research Institute, University Health Network, Toronto, Ontario, Canada

* k.masani@utoronto.ca

## Abstract

Ankle joint quasi-stiffness is a measure representing the control of quiet standing. Here we investigated three methods for determining parameters associated with ankle joint quasi-stiffness. Eleven healthy, young males were asked to stand quietly with eyes open (EO) and eyes closed (EC). Ankle joint quasi-stiffness was computed using two methods, one proposed by Winter et al. (Winter DA, et al. J *Neurophysiol* 80: 1211−1221, 1998) ($K_{qs1}$) and another by Loram and Lakie (Loram ID, et al. J *Physiol* 540: 1111−1124, 2002) ($K_{qs2}$). Both values were normalized using participants' body sizes. The short-term scaling exponent in stabilogram diffusion analysis ($H_s$), which has been linked to quasi-stiffness, was applied. A two-way repeated measures ANOVA revealed a significant main effect of methodology (p < 0.0001, partial η² = 0.42), but no significant effect of eye condition (p = 0.1947, partial η² = 0.06), nor interaction effect (p = 0.4742, partial η² = 0.02). Intraclass Correlation Coefficient (ICC(3,1)) analysis indicates good reliability between $K_{qs1}$ and $K_{qs2}$ with an ICC value of 0.858 (p < 0.001) in the EO condition, and moderate reliability with an ICC value of 0.718 (p = 0.004) in the EC condition. $H_s$ and $K_{qs1}$ exhibited a very strong negative correlation in the EO ($\rho$ = −0.909) condition and a moderately strong negative correlation in the EC ($\rho$ = −0.782) condition. $H_s$ and $K_{qs2}$ exhibited moderately strong negative correlation in both the EO ($\rho$ = −0.664) and EC ($\rho$ = −0.555) conditions. The good and moderate reliability between $K_{qs1}$ and $K_{qs2}$ suggests that both measures capture similar stiffness attributes relating to their neuro-mechanical components, despite a significant methodological difference. Additionally, relatively high correlations of $H_s$ with $K_{qs1}$ and with $K_{qs2}$ suggest that the stochastic characteristics of centre of pressure in the short period indeed reflect the overall quasi-stiffness at the ankle joint.

**Data availability statement:** All relevant data underlying the findings of this study are publicly available in the University of Toronto Dataverse at https://doi.org/10.5683/SP3/KUDTGD.

**Funding:** This work was supported by the Natural Sciences and Engineering Research Council of Canada (NSERC; Grant No. RGPIN-2017-06790). There was no additional external funding received for this study. The funders had no role in study design, data collection and analysis, decision to publish, or preparation of the manuscript.

**Competing interests:** The authors have declared that no competing interests exist.

## Introduction

Human bipedal stance is inherently unstable due to the centre of mass (COM) being high above a relatively small base of support. The ankle joint plays a key role in maintaining COM equilibrium during quiet standing, as it is the primary joint connecting the body to the ground: Specifically, because the COM is in front of the ankle joint, it generates planterflexion torque to counteract gravity pulling the body forward; As a result, the ankle joint torque, which is proportional to the distance between the centre of pressure (COP) and the ankle joint, is highly correlated with the COM displacement [1,2,3], as is plantarflexors' activity [1,2].

Ankle torque is regulated both passively and actively [4]. Passive torque arises from intrinsic joint properties such as stiffness and damping, while active torque is generated by muscle contraction under central nervous system control [4]. Although intrinsic stiffness plays a crucial role, it is insufficient to stabilize quiet standing on its own [5,6,7,4,8]. Active torque compensates for this limitation, enabling adaptive and precise control. While passive torque responds immediately to body kinematics, active torque has inherent delays due to neural transmission and muscle activation [9,7].

Stiffness arising from both passive and neural mechanisms is essential for postural stability. *Quasi-stiffness* [10], a composite measure of both, has been used to assess joint behavior in the hip [11], knee [12,11], and metatarsophalangeal joint [13]. For the ankle during quiet standing, two methods have been proposed [3]. [3] modeled the body as an inverted pendulum with a tuned mass-spring-damper system and calculated quasi-stiffness using the system's undamped natural frequency—referred to here as "*quasi-stiffness 1*" ($K_{qs1}$). [6] decomposed sway into small segments or "unit sways," estimating quasi-stiffness as the slope of the torque-angle plot at equilibrium. We refer to this as "*quasi-stiffness 2*" ($K_{qs2}$).

COP is often analyzed to assess postural control. Stabilogram diffusion analysis (SDA), introduced by [14], has been applied across various populations including those with lower back pain [15], stroke [16], diabetic neuropathy [17], phobic postural vertigo [18], ADHD [19], and Parkinson's disease [20,21], as well as healthy older adults [22,23], obese individuals [24], frail adults [25], and fallers [26,27]. SDA quantifies sway persistence using scaling exponents for short-term (below ~1 s) and long-term (above ~1 s) intervals. A short-term exponent ($H_s$) greater than 0.5 indicates persistent behavior, while values below 0.5 reflect anti-persistence [14]. Studies consistently show that $H_s$ is higher in at-risk individual [22,27,18]. Since ankle quasi-stiffness must act within this short-term interval and intrinsic stiffness alone is insufficient to stabilize sway, we hypothesize that $H_s$ reflects quasi-stiffness during quiet standing.

Therefore, the purposes of this study were twofold in healthy young individuals: (1) to examine the reliability and concurrent validity between the two quasi-stiffness measures ($K_{qs1}$ and $K_{qs2}$, and (2) to determine whether the short-term scaling exponent ($H_s$) is associated with ankle joint quasi-stiffness. We hypothesized that: (i) $K_{qs1}$ and $K_{qs2}$ would demonstrate good reliability and strong correlation, reflecting similar underlying neuro-mechanical stiffness properties despite methodological differences; and (ii) $H_s$ would be negatively correlated with both quasi-stiffness measures, such

that greater short-term persistence (higher $H_s$) would reflect lower effective ankle quasi-stiffness during quiet standing. Given that body rigidity may depend on visual input [28], we examined both eyes-open and eyes-closed conditions.

## Methods

### Participants

Eleven healthy young males (age: 20.7±3.6 years; height: 173.6±7.1 cm; mass: 68.0±8.6 kg; mean ± SD) participated. Data for this study were previously collected by [29]. Participant recruitment was conducted from May 1st to August 31st, 2015. The study received approval from the Research Ethics Board of the University Health Network (Approval No. 12−011), and written informed consent was obtained from all participants.

### Procedure

Participants stood quietly with arms crossed over the chest. Heel distance was set to 11% of height, and feet pointed outward at a 14° angle from the midline [29,30]. Each participant completed 120 seconds of quiet standing with eyes open (EO) and eyes closed (EC), in randomized order.

Body kinematics were recorded using a motion capture system (Rapter-E, Motion Analysis Corp., USA) at 200 Hz. Twenty-nine reflective markers were placed based on the modified Helen-Hayes model [31]. Kinetic data were collected using a dual force plate (AccuSway ACS-DUAL, AMTI, USA) at 2000 Hz.

### Data processing and analysis

Kinematic and kinetic data were processed in MATLAB (2020b, MathWorks, USA) to calculate $K_{qs1}$, $K_{qs2}$, and $H_s$. Analysis focused on the anterior-posterior direction, where quiet standing sway predominantly occurs [3].

Calculation of $K_{qs1}$

Following [3], the horizontal force was band-pass filtered (0.15–4 Hz) using a fourth-order, zero-phase-lag Butterworth filter [32,33,2,34]. COM acceleration ($COM_a$) in the AP direction was computed as:

$$COM_a = \frac{f_h}{m},$$

(1)

where $f_h$ is the filtered horizontal ground reaction force, and m = 0.971·M (M = body mass) [35]. $COM_a$ was segmented into four 30-s intervals, transformed using FFT, and averaged to obtain the amplitude spectrum. A nonlinear least squares fit of a tuned mechanical system was applied:

$$A(\omega) = \frac{C}{\sqrt{1 + [\frac{I\omega}{B} - \frac{K_e}{\omega B}]}},$$

(2)

where $I$, $K_e$, and $B$ are the inertial, spring, and damping constants, and $C$ is a constant. $I$ was determined by anthropometric measure [35]. The undamped natural frequency $\omega_n$ at the spectral peak was used to compute $K_{qs1}$:

$$K_{qs1} = I\omega_n^2 + mGh$$

(3)

where $h$ is the COM height above the ankle and $G$ is the gravitational constant.

Calculation of $K_{qs2}$

Following [6], ankle torque ($T$) and COM angle ($\theta_{COM}$) were decomposed into unit sways—defined as unidirectional segments between zero angular velocity points. Equilibrium was defined when angular velocity was maximal and angular

acceleration was zero. $K_{qs2}$ was defined as the slope of the $T-\theta_{COM}$ curve at equilibrium. Median $K_{qs2}$ was used to represent the trial due to non-normal distribution. Because higher sampling (200 Hz vs. 25 Hz) revealed multiple equilibrium points per sway, data were low-pass filtered at 1 Hz to reduce noise.

*Calculation of $H_s$*

$H_s$ was calculated based on [14]. Unlike their multiple 30-s trials, our data consisted of one 120-s trial. COP displacement in the AP direction was low-pass filtered (5 Hz) [36]. Mean square displacement $< COP_d^2 >$ over time interval $\Delta t$ was calculated as:

$$< COP_d^2 > = \frac{\sum_{i=1}^{N-m} [COP_d(i+m) - COP_d(i)]^2}{N-m},$$

(7)

with N as total data points and m as the number of samples in $\Delta t$. A log–log plot of $< COP_d^2 >$ vs. $\Delta t$ was used to calculate slope estimates. The short-term range (0.08–0.8 s) gave $H_s$; the long-term (2–10 s) gave $H_l$.

## Statistical analyses

Because $K_{qs1}$ and $K_{qs2}$ were body-size dependent, both were normalized using $mGh$ before comparison. Normality was confirmed. Intraclass Correlation Coefficient (ICC(3,1)) analysis was applied to examine the reliability between $K_{qs1}$ and $K_{qs2}$. Bland-Altman plot was also applied to examine the agreement between $K_{qs1}$ and $K_{qs2}$.

Since $H_s$ was non-normally distributed, Spearman correlations were used to examine relationships with $K_{qs1}$ and $K_{qs2}$. A two-way repeated-measures ANOVA assessed effects of eye condition and method on $K_{qs1}$ and $K_{qs2}$, while the Wilcoxon test evaluated eye condition effects on $H_s$. Analyses were performed in JMP 15 (SAS Institute, USA), with $\alpha = 0.05$.

## Results

Fig 1A,1D shows representative raw data from a participant in the EC condition. Fig 1A illustrates the COM acceleration amplitude spectrum used to calculate $K_{qs1}$. The fitted curve of a tuned mechanical circuit showed a high goodness of fit ($R^2 = 0.900$ for this example; group values: EO: 0.788 ± 0.070; EC: 0.830 ± 0.050). Fig 1B displays the angle, angular velocity, and angular acceleration over time, while Figure 1C shows ankle joint torque vs. $\theta_{COM}$, both used in $K_{qs2}$ calculation. As observed by Loram and Lakie, unit sway showed a biphasic drop-and-catch pattern, where the inverted pendulum starts at rest, falls, reaches maximum speed at equilibrium, and returns to rest (inset in Fig 1C). Fig 1D shows a log-log stabilogram-diffusion plot used to calculate $H_s$. $H_s$ and $H_l$ were derived via least-squares estimation and demonstrated high fit ($R^2$: EO short-term: 0.982 ± 0.023; EC: 0.984 ± 0.018; EO long-term: 0.941 ± 0.071; EC: 0.866 ± 0.109).

Fig 2A, 2C presents the distributions of $K_{qs1}$, $K_{qs2}$, and $H_s$. A two-way ANOVA revealed no significant effect of eye condition on $K_{qs1}$ and $K_{qs2}$ (F(1,30) = 1.76, p = 0.1947, partial $\eta^2$ = 0.06), but the effect of methodology was significant (F(1,30) = 21.32, p < 0.0001, partial $\eta^2$ = 0.42). No interaction effects were observed (F(1,30) = 0.53, p = 0.4742, partial $\eta^2$ = 0.02). $H_s$ did not differ significantly between EO and EC (Wilcoxon signed-rank test, p = 0.5771).

Fig 3A–3C (EO) and 3D–F (EC) show relations between the three measures. ICC(3,1) analysis showed good reliability between $K_{qs1}$ and $K_{qs2}$ in EO (ICC = 0.857, F(10,10) = 13.038, p = 0.00018, 95% CI: [0.56, 0.96]) and moderate reliability in EC (ICC = 0.718, F(10,10) = 6.101, p = 0.00425, 95% CI: [0.24, 0.92]).

Bland–Altman plots (Figs 4A–4B) showed symmetrical distributions and consistent differences between $K_{qs1}$ and $K_{qs2}$, with $K_{qs1}$ being higher in both EO (mean difference: 0.31) and EC (0.23).

Spearman correlations showed a very strong negative correlation between $H_s$ and $K_{qs1}$ in EO ($\rho = -0.909$), and a moderately strong correlation in EC ($\rho = -0.782$). For $H_s$ and $K_{qs2}$, correlations were moderately strong in both EO ($\rho = -0.664$) and EC ($\rho = -0.555$).

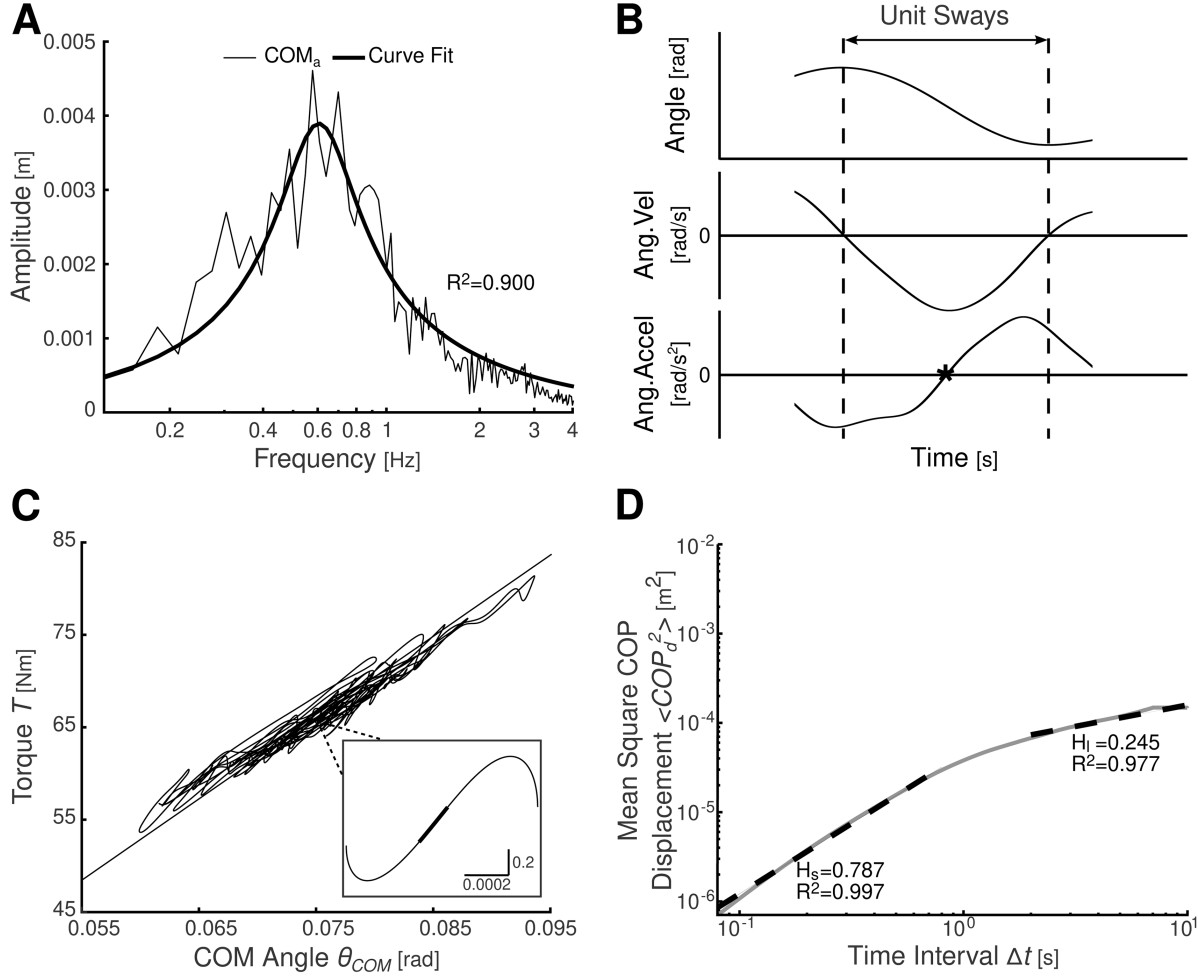

**Fig 1. Plots of raw-data resultant plots for participants with the EC condition. (A)** Amplitude spectrum of the **$COM_a$** plotted on a log scale. **$K_{qs1}$** is determined from the $\omega_n$ of the system which corresponds to where the peak of the spectrum occurs. **(B)** Decomposition of postural sway into unit sways. A unit sway is defined as a unidirectional sway from one reversal point (i.e., angular velocity is zero) to another. The equilibrium occurs at when the angular acceleration is zero and this is identified on the plots with the *. **(C)** Plot of **T** against $\theta_{COM}$ for the entire trial. The straight line indicates the line of equilibrium, or the gravitational torque on the pendulum. The inset plot is of **T** against $\theta_{COM}$ for a unit sway. The thicker line indicates where the slope was computed to find **$K_{qs2}$** **(D)** Log-log of stabilogram-diffusion plot with fitted regression lines. **$H_l$** and **$H_s$** values, as well as the coefficient of determination, **$R^2$**, values are shown.

## Discussion

Our first aim was to compare two methods of calculating ankle joint quasi-stiffness proposed by [3] ($K_{qs1}$) and [6] ($K_{qs2}$). We found good (EO) and moderate (EC) reliability based on ICC analysis, with $K_{qs1}$ consistently larger than $K_{qs2}$. The second aim was to investigate whether the short-term scaling exponent ($H_s$) obtained from stabilogram diffusion analysis relates to either $K_{qs1}$ or $K_{qs2}$. $H_s$ showed very or moderately strong negative correlations with both, in both eye conditions. Since both $K_{qs1}$ and $K_{qs2}$ are based on the single-link inverted pendulum model, and as its dynamics differ depending on eye condition [37,28], we expected potential effects from visual input. However, no significant effects were observed in any analyses.

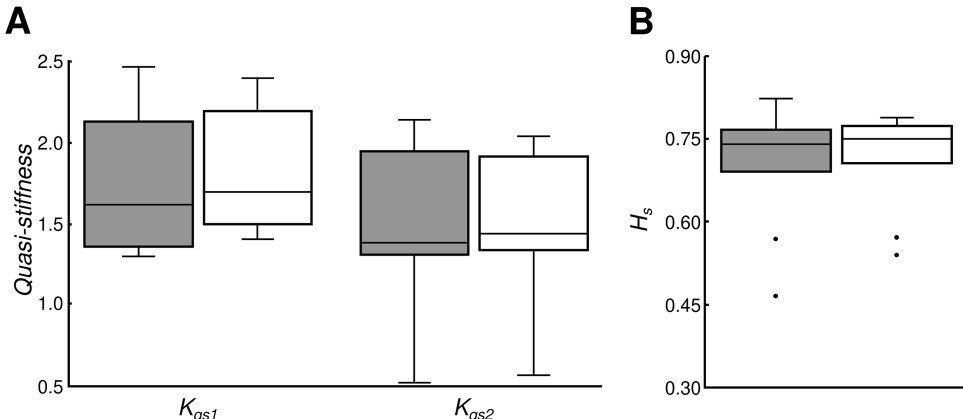

**Fig 2. Distribution of calculated (A) intrinsic stiffness, $K_{qs1}$, and quasi-stiffness, $K_{qs2}$, as well as (B) short-term scaling exponent, $H_s$, values.** EO are shaded in grey, while EC are not shaded.

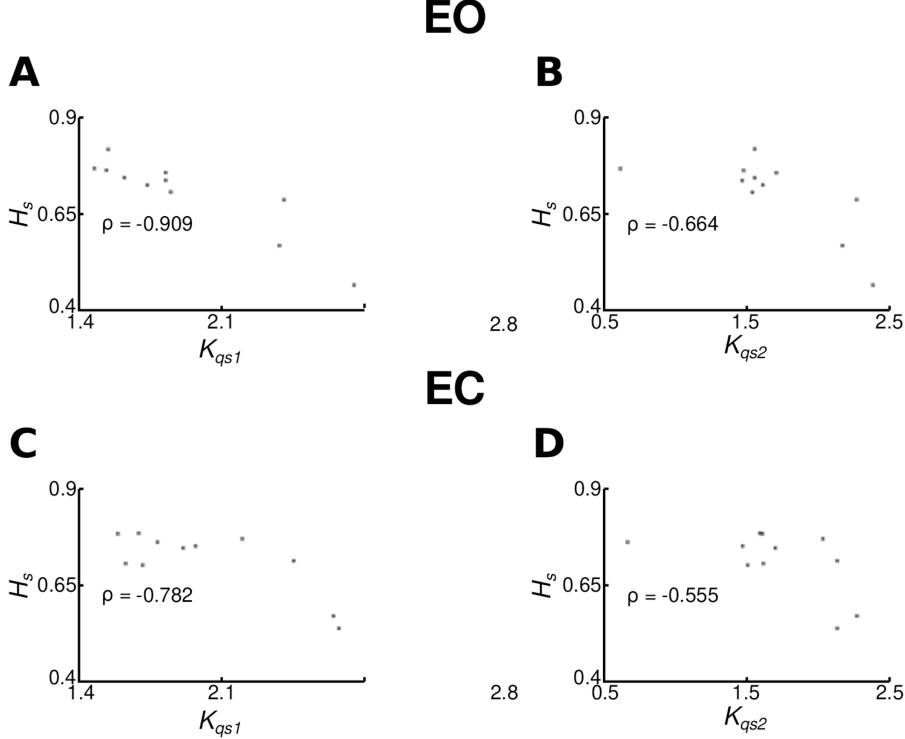

**Fig 3. Correlation plots and the Spearman correlation coefficients between (A) $K_{qs1}$ and $K_{qs2}$, (B) $H_s$ and $K_{qs1}$ and (C) $H_s$ and $K_{qs2}$ for the EO condition.** Plots (D), (E) and (F) are for the EC condition.

## Relation between $K_{qs1}$ and $K_{qs2}$

We observed good/moderate reliability between $K_{qs1}$ and $K_{qs2}$ values. This was expected since both these values theoretically represent the same metric, which has never been compared in the past. However, there are several differences

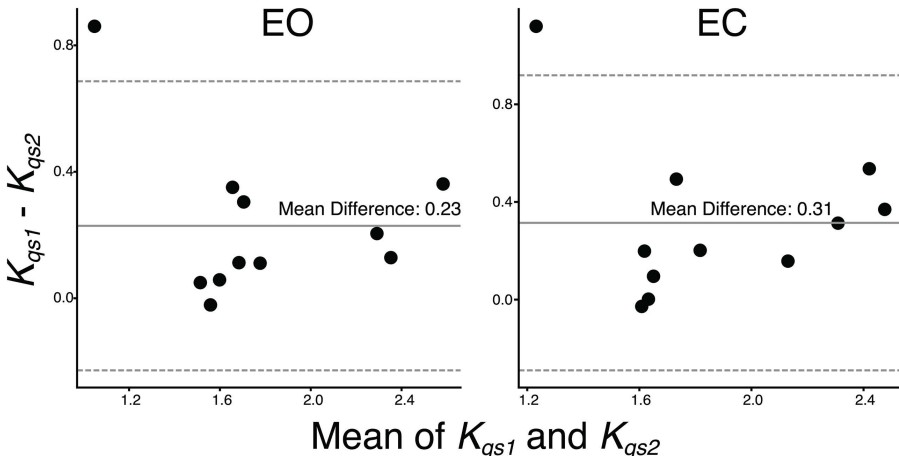

**Fig 4. Bland–Altman plots comparing quasi-stiffness values calculated by $K_{qs1}$ and $K_{qs2}$ in both EO and EC conditions.** The left panel shows the plot for the EO condition, and the right panel for the EC condition. The mean differences (solid lines) and 95% limits of agreement (dashed lines) are indicated.

in these measures. $K_{qs1}$ is an estimate of quasi-stiffness calculated based on the single-linked inverted pendulum model with a tuned mass-spring-damper system. A $K_{qs1}$ value is calculated using a whole time series of COP corresponding to the average quasi-stiffness for the entire recorded period. On the contrary, $K_{qs2}$ is a direct measure of the quasi-stiffness, which is measured as the slope of the torque-angle plot for each unit sway. As quiet standing consists of number of unit sways, multiple $K_{qs2}$ values can be obtained during a quiet standing task, which forms a diverse distribution of samples. These differences may cause the consistent difference between the two variables. In addition, $K_{qs2}$ is more sensitive to signal fluctuations and the selected filter parameters, since it depends on precise detection of zero-crossings and slope estimation for short-duration events. $K_{qs1}$, which is based on the overall spectral profile, is less affected by filtering choices. Despite this, the strong correlation observed between the two supports the robustness of $K_{qs2}$ under our processing conditions.

Quasi-stiffness is a measure reflecting how the ankle joint torque is exerted to control the COM displacement. Hence, we can expect that quasi-stiffness can be used to evaluate the postural control system. For example, [38]. [38] demonstrated that $K_{qs1}$ increases with facing postural threat induced by increased floor height. The postural threat affects the postural control strategy which is shown in the quasi-stiffness. $K_{qs1}$ is a convenient measure to be used as it only requires a force plate compared to $K_{qs2}$ which requires measuring the body's kinematics. On the contrary, $K_{qs1}$ only provides the average quasi-stiffness, which dynamically changes moment-to-moment, a characteristic that can be evaluated using $K_{qs2}$. The current study suggests that both methods may provide useful indicators of postural control, although each has its own advantages and limitations.

The group average values of $K_{qs1}$ and $K_{qs2}$ were significantly different (Fig 2), with $K_{qs1}$ consistently larger than $K_{qs2}$. One possible contributor to this difference is that $K_{qs2}$ yields a distribution of values across unit sways, and we summarized this distribution using the median. Because $K_{qs1}$ represents a single global estimate derived from the entire time series, differences in statistical summarization may partially influence the observed bias.

However, rather than attributing this difference solely to this statistical aspect, it is important to consider how stiffness is estimated in each method. $K_{qs2}$ is derived from the local slope of the torque–angle relationship specifically at equilibrium points during individual unit sways. These equilibrium points may not necessarily reflect the overall restoring characteristics governing the entire sway cycle. In contrast, $K_{qs1}$ is inferred from the global oscillatory behavior of the inverted

pendulum model using the entire time series. As such, it reflects the effective restoring stiffness required to account for the observed sway dynamics across time. If the torque–angle relationship varies throughout the sway cycle, the local slope sampled at equilibrium may differ from the effective stiffness that determines the system's oscillatory behavior. This difference in temporal scale and signal utilization may contribute to the systematically larger magnitude of $K_{qs1}$ compared to $K_{qs2}$.

Additionally, the normalization applied to $K_{qs1}$ and $K_{qs2}$ using $mgh$ introduces a shared constant, which could potentially inflate their correlation. However, the similarly strong correlations observed in the absolute (non-normalized) values (EO r = 0.845; EC r = 0.934) suggest that the relationship between the two measures is not merely an artifact of normalization.

## Relation between $K_{qs1}$/$K_{qs2}$ and $H_s$

We found a high negative correlation both between $H_s$ and $K_{qs1}$ as well as between $H_s$ and $K_{qs2}$. All $H_s$ values were larger than 0.5 which indicate persistence behaviour in the short term, i.e., less than 1 s. The dominant oscillation in $COM_a$ is approximate 0.5 Hz (Fig 1A) giving 2 s for a cycle that consists of two unit sways. One unit sway is about 1 s in its duration [6]. Therefore, all of $H_s$, $K_{qs1}$ and $K_{qs2}$ reflect the dynamics of postural sway in 1 s. Within 1 s, a fall due to the insufficient intrinsic stiffness [9,7] occurs initiating a unit sway, while active torque captures the falling body. The fall due to the insufficient intrinsic stiffness can cause persistent behaviour in $H_s$, while the whole dynamics including the active torque behaviour can be reflected in $K_{qs1}$ and $K_{qs2}$.

This mechanism may account for the high correlation among the three quasi-stiffness parameters, and it also supports the direction of the observed correlations. Specifically, a negative correlation between $Hs$ and the quasi-stiffness measures is expected because greater persistence (i.e., higher $Hs$) reflects less mechanical resistance to perturbation, or lower stiffness. This interpretation is supported by prior studies showing elevated $Hs$ values in at-risk elderly individuals and neurological populations, such as Parkinson's disease and stroke [22,20,16]. In the present study, higher $Hs$ was associated with lower ankle quasi-stiffness. Taken together, these findings suggest that populations exhibiting elevated $Hs$ may also demonstrate reduced ankle quasi-stiffness during quiet standing. However, this relationship has not been directly verified in such clinical groups.

Future studies should therefore concurrently quantify $Hs$ and ankle quasi-stiffness in elderly and neurological populations to determine whether the inverse association observed in healthy young individuals is preserved under pathological conditions. If confirmed, $Hs$, which is obtainable from force plate measurements alone, may serve as a clinically feasible surrogate biomarker of ankle quasi-stiffness regulation without requiring detailed kinematic assessment.

## Study limitation

This study utilized previously collected data from Fok et al. [29], which limited our ability to perform an a priori estimation of the required sample size. Although the sample size was relatively small ($n$ = 11), the large correlations observed between $Hs$ and the quasi-stiffness measures, as well as the moderate-to-large effect size for methodological differences (partial $\eta^2$ = 0.42), suggest that the primary findings are unlikely to be attributable to sampling variability alone. However, the small effect sizes observed for eye condition ($\eta^2$ = 0.06) and interaction ($\eta^2$ = 0.02) indicate limited sensitivity to detect subtle effects, and such findings should therefore be interpreted cautiously.

Additionally, the participant characteristics were exclusively young, healthy male individuals, restricting the generalizability of our findings to females, older adults, and clinical populations.

## Conclusion

In conclusion, we demonstrated that $K_{qs1}$ and $K_{qs2}$ are strongly correlated and exhibit good to moderate reliability, suggesting that they reflect similar stiffness-related characteristics of postural control, despite some systematic differences in values. Additionally, $H_s$ was negatively correlated with both $K_{qs1}$ and $K_{qs2}$, supporting its relevance as an indirect indicator

of ankle joint quasi-stiffness during quiet standing. These relationships were consistent regardless of visual input. While these measures appear to capture related aspects of postural control, they are not interchangeable. Their methodological differences, along with the limitations of the current sample (i.e., healthy young males), warrant further research to evaluate their comparability across different populations and contexts [39,40,41,42,43,44,45,46,47].

## Acknowledgments

The authors would like to acknowledge Dr. Albert Vette for his assistance with the $K_{qs2}$ analysis.

## Declaration of generative AI and AI-assisted technologies in the writing process

During the preparation of this work the authors used ChatGPT in order to enhance the clarity and quality of the writing. After using this tool/service, the authors reviewed and edited the content as needed and take full responsibility for the content of the publication.

## Author contributions

**Conceptualization:** Kei Masani.

**Data curation:** Kaylie Lau, Kai Lon Fok, Jonguk Lee.

**Formal analysis:** Kaylie Lau, Kai Lon Fok, Kei Masani.

**Funding acquisition:** Kei Masani.

**Investigation:** Kai Lon Fok, Kei Masani.

**Methodology:** Kaylie Lau, Kai Lon Fok, Jonguk Lee, Kei Masani.

**Project administration:** Kai Lon Fok.

**Resources:** Kei Masani.

**Software:** Kei Masani.

**Supervision:** Kei Masani.

**Validation:** Kai Lon Fok, Kei Masani.

**Visualization:** Kai Lon Fok, Kei Masani.

**Writing – original draft:** Kaylie Lau.

**Writing – review & editing:** Kaylie Lau, Kai Lon Fok, Jonguk Lee, Kei Masani.

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
