## [Decision Letter · Decision Letter 0]

11 Feb 2026

Dear Dr. Masani,

Thank you for submitting your manuscript to PLOS ONE. After careful consideration, we feel that it has merit but does not fully meet PLOS ONE’s publication criteria as it currently stands. Therefore, we invite you to submit a revised version of the manuscript that addresses the points raised during the review process.

We look forward to receiving your revised manuscript.

Kind regards,

Seyed Hamed Mousavi

Academic Editor

PLOS One

Journal Requirements:

https://journals.plos.org/plosone/s/file?id=ba62/PLOSOne_formatting_sample_title_authors_affiliations.pdf....

“This work was supported by a grant from the Natural Sciences and Engineering Research Council of Canada (Grant No. RGPIN- 2017–06790).”

“This work was supported by a grant from the Natural Sciences and Engineering Research Council of Canada (Grant No. RGPIN- 2017–06790).”

5. In the online submission form you indicate that your data is not available for proprietary reasons and have provided a contact point for accessing this data. Please note that your current contact point is a co-author on this manuscript. According to our Data Policy, the contact point must not be an author on the manuscript and must be an institutional contact, ideally not an individual. Please revise your data statement to a non-author institutional point of contact, such as a data access or ethics committee, and send this to us via return email. Please also include contact information for the third party organization, and please include the full citation of where the data can be found.

Reviewers' comments:

Reviewer's Responses to Questions

**Comments to the Author**

1. Is the manuscript technically sound, and do the data support the conclusions?

Reviewer #1: Yes

Reviewer #2: Yes

2. Has the statistical analysis been performed appropriately and rigorously?

Reviewer #1: No

Reviewer #2: Yes

3. Have the authors made all data underlying the findings in their manuscript fully available?

Reviewer #1: Yes

Reviewer #2: Yes

4. Is the manuscript presented in an intelligible fashion and written in standard English?

Reviewer #1: Yes

Reviewer #2: Yes

Reviewer #1: This study was aimed to investigate the relationship across 1, 2, and in healthy young individuals. The results showed that the good and moderate reliability between 1 and 2 suggests that both measures capture similar stiffness attributes relating to their neuro-mechanical components. Additionally, relatively high correlations of with 1 and with 2 suggest that the stochastic characteristics of center of pressure in the short period indeed reflect the overall quasi-stiffness at the ankle joint. Overall, the study is interesting, however there are some clarifications needed.

Comment#1

Introduction: please state hypothesis/hypotheses for this study.

Comment#2

Abstract, please insert effect size for repeated measure ANOVA performed.

Reviewer #2: Dear Corresponding Author,

Your article titled "Comparison of methods for determining parameters related to ankle joint quasi-stiffness during quiet standing" is a valuable contribution to the field of biomechanics and postural control because of the unique perspective it offers. To make the scientific impact of your work as strong as possible, I suggest the following improvements.

- First, it would be useful to have a deeper discussion about the mechanical and modeling reasons for the systematic difference (bias) found between the methods, using current literature to explain these findings.

- Second, I believe the limitations section needs to be more transparent about how the sample size affects the statistical power of your results. Since the study is limited to 11 healthy young male participants , it is difficult to apply these findings to other groups, and the lack of female participants and the narrow age range are significant weaknesses.

- Finally, adding a paragraph on how the relationship between the short-term scaling exponent and quasi-stiffness might look in different clinical groups, such as the elderly or people with neurological disorders, would greatly improve the clinical value and future research potential of your work.

.

Reviewer #1: No

Reviewer #2: **Yes:**Gizem Irem KINIKLI, Prof., PhD., PT.Gizem Irem KINIKLI, Prof., PhD., PT.Gizem Irem KINIKLI, Prof., PhD., PT.Gizem Irem KINIKLI, Prof., PhD., PT.

---

## [Author Response · Author response to Decision Letter 1]

19 Feb 2026

We thank the editor and reviewers for their constructive and insightful comments. In response, we have revised the manuscript to improve clarity, statistical transparency, and discussion depth. Specifically, we have (1) clarified the study hypotheses in the Introduction, (2) reported effect sizes for the repeated-measures ANOVA, (3) expanded the mechanistic discussion regarding differences between quasi-stiffness estimation methods, (4) elaborated on the clinical implications of the relationship between H_s and quasi-stiffness, and (5) strengthened the Study Limitation section to more explicitly address statistical power and generalizability.

Detailed, point-by-point responses to all reviewer comments are provided in the attached Response Letter.

---

## [Editor Report · Decision Letter 1]

20 Mar 2026

Comparison of methods for determining parameters related to ankle joint quasi-stiffness during quiet standing

PONE-D-25-61533R1

Dear Dr. Masani,

We’re pleased to inform you that your manuscript has been judged scientifically suitable for publication and will be formally accepted for publication once it meets all outstanding technical requirements.

Kind regards,

Seyed Hamed Mousavi

Academic Editor

PLOS One
---

## [Editor Report · Acceptance letter]

PONE-D-25-61533R1

PLOS One

Dear Dr. Masani,

I'm pleased to inform you that your manuscript has been deemed suitable for publication in PLOS One. Congratulations! Your manuscript is now being handed over to our production team.

Kind regards,

on behalf of

Dr. Seyed Hamed Mousavi

Academic Editor

PLOS One